# Improved Calculation of Nonlinear Near-Bed Wave Orbital Velocity in Shallow Water: Validation against Laboratory and Field Data

**Pham Thanh Nam** [1,2,*] **, Joanna Staneva** [1] **, Nguyen Thi Thao** [1,2] **and Magnus Larson** [3]

[1]  Department of Hydrodynamics and Data Assimilation, Centre for Materials and Coastal Research, Helmholtz-Zentrum Geesthacht, 21502 Geesthacht, Germany; joanna.staneva@hzg.de (J.S.); thi.nguyen@hzg.de (N.T.T.)

[2]  Institute of Mechanics, Vietnam Academy of Science and Technology (VAST), 100000 Hanoi, Vietnam; ptnam@imech.vast.vn (P.T.N.); ntthao@imech.vast.vn (N.T.T.)

[3]  Water Resources Engineering, Lund University, S-22100 Lund, Sweden; magnus.larson@tvrl.lth.se

*  Correspondence: nam.pham@hzg.de; Tel.: +49-4152-87-1884

**Abstract:** A new parameterization for calculating the nonlinear near-bed wave orbital velocity in the shallow water was presented. The equations proposed by Isobe and Horikawa (1982) were modified in order to achieve more accurate predictions of the peak orbital velocities. Based on field data from Egmond Beach in the Netherlands, the correction coefficient and maximum skewness were determined as functions of the Ursell number. The obtained equations were validated against measurements from Egmond Beach, and with laboratory data from small-scale wave flume experiments at Delft University of Technology and from large-scale wave flume experiments at Delft Hydraulics. Inter-comparisons with other previously developed parameterizations were also carried out. The model simulations by the present study were in good agreement with the measurements and have been improved compared to the previous ones. For Egmond Beach, the root-mean-square errors for the peak onshore ($u_c$) and offshore ($u_t$) orbital velocities were approximately 21%. The relative biases were small, approximately 0.013 for $u_c$ and $-0.068$ for $u_t$. The coefficient of determination was in the range between 0.64 and 0.68. For laboratory experiments, the root-mean-square errors in a range of 7.2%–24% for $u_c$, and 7.9%–15% for $u_t$.

**Keywords:** velocity skewness; velocity asymmetry; orbital velocity; wave non-linearity; sediment transport

## 1. Introduction

Velocity skewness plays an important role in sediment transport and beach morphological change in shallow water, especially for the cross-shore transport. The difference between the onshore and offshore velocities during a wave cycle [1] often generates onshore-directed sediment transport [2,3] and the net transport induced by velocity skewness and undertow are main factors in forming nearshore bars in shallow water [4–6]. During calm weather conditions, velocity skewness is also considered a key factor for beach recovery after a storm [6,7]. Recently, Albernaz et al. [8] investigated the effects of wave orbital velocity parameterization on the sediment transport and long-term morphodynamics, and their results showed that a small difference in the calculation of wave orbital velocities and velocity skewness could cause large uncertainty in predicting long-term morphodynamics in the nearshore. Therefore, accurate estimates of the nonlinear near-bed orbital velocities in shallow water are required and essential to calculate sediment transport and morphological change in shallow water.

The pioneering work of Stokes [9] demonstrated that the onshore part of a wave becomes shorter with a higher crest and the offshore part longer with a lower trough when the wave propagates in decreasing water depth. Cornish [10] observed that the movement of coarser sediments due to shoreward velocities associated with the wave crest was more pronounced than the movement due to seaward velocities associated with the wave trough. Van de Werf et al. [11] introduced a database of approximately 300 experiments carried out under full-scale non-breaking waves and non-breaking waves, also in combination with a mean current, in which most of the experiments involved velocity skewed flows, resulting in a net transport shoreward. The horizontal asymmetry of the wave shape produces the skewed wave orbital velocities, and the term defined by the ratio $u_c/(u_c + u_t)$, where $u_c$ and $u_t$ are the near-bed peak onshore and offshore orbital velocities, respectively, is commonly referred to as velocity skewness [2].

Strong asymmetry often occurs in very shallow water, especially in the swash zone. From the field experiment conducted on Barret Beach, Fire Island, New York, Conley and Griffin [12] showed the swash flow during the uprush phase caused approximately a double bed shear stress compared to the backwash phase, while the backwash duration was 135% longer than that of the uprush. Miles et al. [13] investigated the sediment transport processes in the swash zone of a dissipative beach and a steeper beach in the south west of England based on high-frequency measurements of water depth, current velocity, and suspended sediment concentration. A net shoreward transport often attained a maximum in the mid swash zone for both beaches, indicating sediment deposition in the shoreward half of the swash. A net transport in the swash zone was also observed in the shoreward part for almost all laboratory experiments in the flume of the Polytechnic University of Catalonia (UPC), although the beach profiles were eroded under the erosive conditions [14]. In addition, Silva et al. [15] evaluated the net transport rates under sheet flow conditions and concluded that the velocity skewness increases the magnitude of the net onshore transport rates.

There have been a number of studies dealing with velocity skewness. Several numerical models based on the Boussinesq equations were developed that could reasonably well reproduce measurements of time-varying orbital velocities and velocity skewness through the shoaling region. For example, Elgar et al. [16] employed a Boussinesq model for the nonbreaking unidirectional gravity waves. The model was validated against data sets from two natural beaches at Torrey Pines and Santa Barbara and the obtained results of surface elevation, near-bed horizontal orbital velocity and acceleration were in good with measurements. However, predictions of velocity skewness at breaking and in the surf zone were significantly different from measurements [17,18]. In some cases, especially for highly non-linear waves, large discrepancies between measurements and calculations of skewness by the Boussinesq model have been obtained [19]. Furthermore, such Boussinesq models require fine resolution in both time and space; therefore, their applications are often only suitable for small coastal areas and short-term simulations.

Semi-empirical formulations have been widely used for determining the velocity skewness. Isobe and Horikawa [1] introduced a set of expressions for calculating the near-bed peak onshore and offshore orbital velocities using a combination of fifth-order Stokes and third-order cnoidal wave theories, in which two important parameters including (i) the correction coefficient was a function of the offshore wave height ($H_0$) and wavelength ($L_0$), and (ii) the maximum skewness was determined based on beach slope ($\beta$). Hamm [20] tested the formula of [1] against several data sets from the laboratory. The obtained results showed that the method can well reproduce measurements of the near-bed orbital velocity outside the surf zone. However, the calculated near-bed orbital velocity was not good inside the surf zone, especially for monochromatic waves.

Grasmeijer and Ruessink [21] modified the approach of [1] to determine the onshore and offshore orbital velocities. Their study showed the impact of beach slope on the maximum skewness was small and could be neglected. The maximum skewness was calculated based on the water depth ($d$) and wavelength ($L$; see Table 1). The correction parameter was also modified, depending on the local significant wave height ($H_s$) and water depth. The predictions by the modified formula were in good

agreement with measurements for some laboratory data sets, although the calculated orbital velocities were overestimated landward of the breakpoint for the field data sets.

**Table 1.** Important parameters in the Isobe and Horikawa [1] formulas.

| Parameters | Original Formulas | Modified Formulas of [21] |
|:---:|:---:|:---:|
| Correction parameter | $r_1 - r_2 \exp\{-r_3(d/L_0)\}$ [1] | $1 - 0.4\,(H_s/d)$ |
| Maximum skewness | $0.62 + 0.003/\beta$ | $-2.5d/L + 0.85$ |

[1] $r_1 = 1,\ r_2 = 3.2(H_0/L_0),\ r_3 = -27\log_{10}(H_0/L_0) - 17$.

Doering et al. [6] parameterized velocity skewness based on the local wave parameters, including significant wave height, wavelength, and mean water depth. The predicted skewness agreed well with the measured skewness for data sets from Terschelling and Sandyduck97 sites. Nevertheless, for the data from the Egmond and Duck sites, a large amount of scatter between calculated and measured skewness was obtained.

Elfrink et al. [22] introduced another formula to compute the time-varying orbital velocity. This empirical expression was validated with several field data. For short periods of simulation, the calculations of the orbital velocity were satisfactory. However, the variation between measured and calculated velocity skewness was large; the calculated skewness by [22] often underestimated the measurements. The study of Broek [23] also showed that the skewness based on [22] generally underestimates measurements when tested against several laboratory data sets.

Abreu et al. [24] derived an analytical formula for the near-bed wave orbital velocity. However, it is only applicable for the monochromatic wave. Ruessink et al. [25] parameterized the non-linear coefficient and the phase in the expression of [24] in order to generalize it to random waves. However, this parameterization often overpredicts the onshore velocity and underpredicts the offshore velocity compared with measurements. Rocha et al. [26] modified the aforementioned method using the beach slope and offshore wave conditions in parameterization with regard to nonlinearity and the phase. Although the obtained results were improved, their comparisons and validation were only taken based on a few experimental runs in a small-scale wave flume at mild slopes.

The main aim of this paper is to revise parameterizations in order to improve the calculation of velocity skewness in shallow water. In order to effectively and accurately predict the peak onshore and offshore orbital velocities in the shallow water, the formula of Isobe and Horikawa [1] was modified and parameterized based on field measurements collected at Egmond Beach in the Netherlands [27,28]. Specifically, the correction coefficient and maximum skewness in the formulas were determined based on the Ursell number, producing more reasonable results for the peak onshore and offshore orbital velocities.

The model was validated against the field data at Egmond beach [27,28] and several data sets collected from small-scale laboratory experiments at Delft University of Technology [29] and large-scale laboratory experiments in the Delta flume of Delft Hydraulics [30].

The structure of the paper is as follows: Section 2 briefly introduces the employed data collected at Egmond Beach used for the velocity skewness parameterization and several data sets from the small and large-scale laboratory [29,30]. Then, the nearshore random wave transformation and the modified formulas for nonlinear near-bed orbital velocity, are provided in Section 3. After that, the model validations with laboratory and field data are presented in Section 4. Some points related to the results are discussed in Section 5 and finally concluding remarks are given.

## 2. Employed Data

A field campaign was carried out during approximately six weeks in October–November 1998 at a location 1 km south of Egmond ann Zee in the Netherlands. The beach topography was characterized by two parallel sand bars [27,28]. The offshore wave conditions were measured with a wave buoy in a water depth of 16 m, approximately 5 km offshore, and a wide range of wave conditions were observed.

During the first 10 days of the campaign (15–24 October 1998), the offshore significant wave height varied from 1 to 3 m. The next 6 days (25–30 October 1998) encompassed the most energetic period, and very large offshore waves were observed, where the significant wave height often exceeded 2 m and reached a maximum value of 5.1 m (Figure 1a). Then, the offshore waves were suddenly less energetic, but shortly those conditions were interrupted by a storm (6 Nov. 1998) with the maximum offshore significant wave of around 4 m. After this event, the offshore significant wave height was typically less than 1.5 m. The significant wave period varied from 3.8 to 10.5 s, and the incident wave angles mainly between −50° and +50°, relative to shore normal (Figure 1b,c, respectively). The tidal range varied from 1.4 to 2.1 m (Figure 1d).

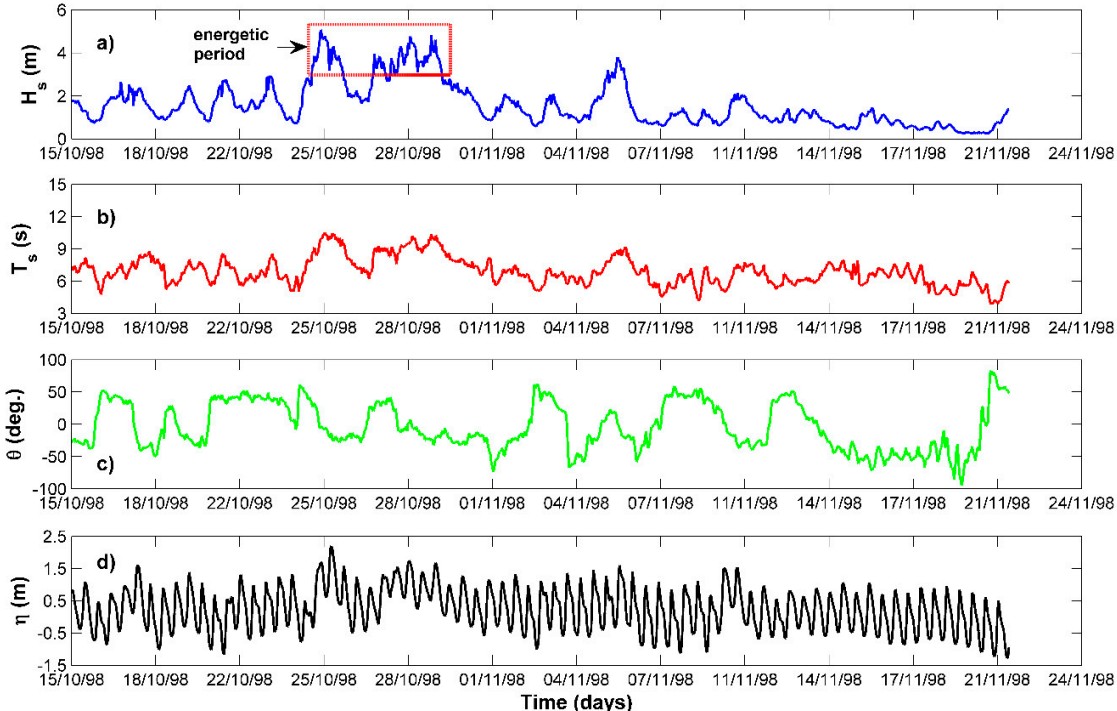

**Figure 1.** Offshore significant wave height (**a**); wave period (**b**); wave direction (**c**); and tidal level; (**d**) recorded during field measurements at Egmond ann Zee in the Netherlands.

The nearshore wave parameters were measured by pressure sensors at six locations of which five were in the vicinity of the inner bar (from E1 to E5), and the remaining location (E6) was seaward of the outer bar (see Figure 2). The measurements of the peak onshore and offshore velocities were observed by bidirectional electromagnetic flow (EMF) meters at a nominal height between 0.2 and 0.5 m above the bed at five locations, E1–E6 [25,27,28]. Data were measured approximately 34 min/hour at a sampling rate of 2 or 4 Hz. These measurements were employed in the present study to determine the correction coefficient and maximum skewness in the formulas of [1], to validate the modified formulas, and to compare the obtained results with previous formulas.

Furthermore, the model was validated against two data sets (B1 and B2) collected from small-scale laboratory experiments at Delft University of Technology [29], called GR99 data hereafter, and three data sets (Test 1A–C) from a large-scale laboratory experiment, LIP11D, in the Delta flume of Delft Hydraulics [30]. Table 2 summarized the collected data at Egmond Beach and at small- and large-scale wave flumes for model validation.

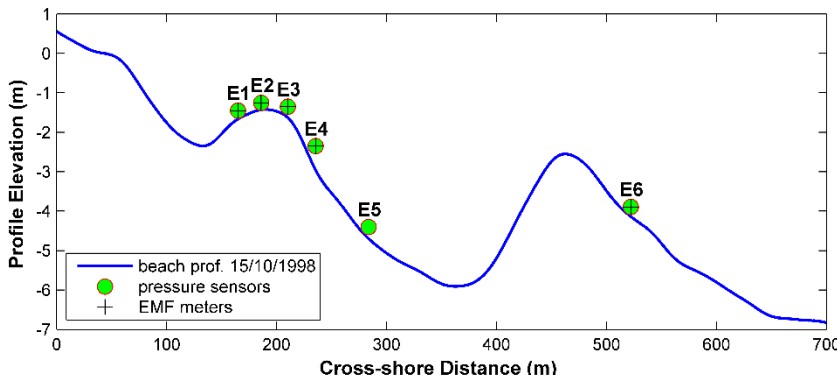

**Figure 2.** Beach topography and measured locations at Egmond Beach, the Netherlands.

**Table 2.** Offshore wave conditions and measured data for model validation.

| Data | $H_{mo}$ (m) | $T_p$ (s) | $\theta$ (deg.) | Parameters | Beach Profile |
|---|---|---|---|---|---|
| Egmond [27,28] | 0.25–5.1 | 3.87–10.77 | −50–50 | $H_s, u_c, u_t$ | Barred beach |
| GR99 [29] | 0.16, 0.19 | 2.3 | 0 | $H_s, u_c, u_t$ | Barred beach |
| LIP11D [30] | 0.9, 1.4, 0.6 | 5 | 0 | $H_s, u_c, u_t$ | Equilibrium |

## 3. Model Description

### 3.1. Nearshore Wave Transformation

In order to calculate the near-bed peak onshore and offshore orbital velocities, as well as velocity skewness, the wave parameters are needed and they were calculated by using the nearshore random wave transformation model denoted Modified-EBED [31–33]. The model is based on the wave energy balance equation as,

$$\frac{\partial(v_x S)}{\partial x} + \frac{\partial(v_y S)}{\partial y} + \frac{\partial(v_\theta S)}{\partial \theta} = \frac{\kappa}{2\omega}\left\{\left(CC_g \cos^2\theta\, S_y\right)_y - \frac{1}{2}CC_g \cos^2\theta\, S_{yy}\right\} - \frac{K}{h}C_g S\left\{1 - \left(\frac{\Gamma h}{H_s}\right)^2\right\}, \quad (1)$$

where $S$ = angular-frequency spectrum density, $(x, y)$ = horizontal coordinates, $\theta$ = angle measured counterclockwise from the x axis, $\kappa$ = free parameter, $\omega$ = wave frequency, $C$ = phase speed, and $C_g$ = group speed, $(v_x, v_y, v_\theta)$ = propagation velocities in their respective coordinate directions, $h$ = still water depth, $K$ = decay coefficient, and $\Gamma$ = stable coefficient.

The output of the model includes three main parameters: significant wave height, significant wave period, and mean wave direction, determined from the wave spectrum. The nearshore wave transformation was validated against various laboratory and field data sets, and the obtained results were in very good agreement with measurements [32,33].

### 3.2. Nonlinear Near-Bed Orbital Velocity

The original method of Isobe and Horikawa [1] was based on fifth-order Stokes wave theory and third-order cnoidal wave theory. In this formula, the full amplitude of the near-bed orbital velocity (Figure 3) is derived as,

$$\hat{u} = 2rU_w, \quad (2)$$

where $U_w$ = near-bed horizontal orbital velocity amplitude using linear wave theory [34] and $r$ = correction coefficient that can be determined based on laboratory and field data.

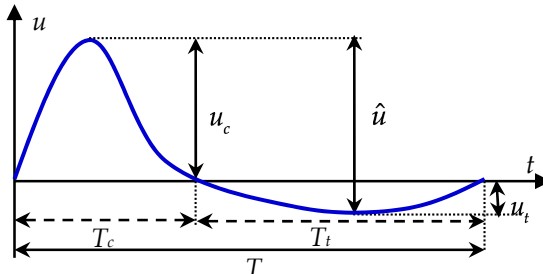

**Figure 3.** Definitions of variables for an asymmetric velocity profile.

In the present study, based on the field data collected at the five locations (E1, E2, E3, E4, and E6) on Egmond Beach [25,27,28], the correction coefficient is dependent on the Ursell number (Figure 4) according to the following fitting function

$$r = p_1 \ln(U_r) + p_2, \tag{3}$$

where $U_r$ = Ursell number, which is determined [35] as,

$$U_r = \frac{H_s L^2}{d^3}. \tag{4}$$

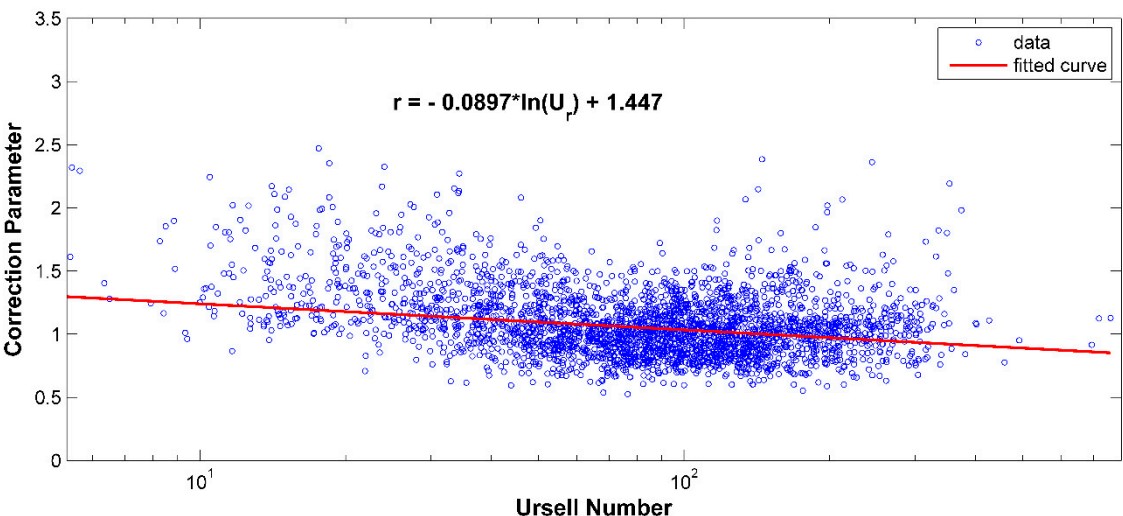

**Figure 4.** Relationship between the correction coefficient and the Ursell number.

Note that Equation (4), used to determine the Ursell number, is different from the equation employed by Ruessink et al. [25]. In their study, the Ursell number was calculated as $U_r = \frac{3}{8} \frac{H_s k}{(kd)^3} = \frac{3}{32\pi^2} \frac{H_s L^2}{d^3}$, with $k$ is local wave number (= $2\pi/L$). Therefore, the value of the Ursell number using Equation (4) in our study is greater than the one by [25], in proportion $\frac{32\pi^2}{3} : 1$.

Using linear least-square fitting, the best fit coefficient values were $p_1 = -0.0897 \pm 0.0038$, and $p_2 = 1.447 \pm 0.017$ with ranges representing the 95% confidence limits.

The near-bed peak onshore orbital velocity ($u_c$) is calculated following the approach of [1] as,

$$\left(\frac{u_c}{\hat{u}}\right) = 0.5 + \left(\left(\frac{u_c}{\hat{u}}\right)_{\max} - 0.5\right) \tanh\left(\frac{(u_c/\hat{u})_a - 0.5}{(u_c/\hat{u})_{\max} - 0.5}\right), \tag{5}$$

where $\left(\frac{u_c}{\hat{u}}\right)_{max}$ is maximum skewness and $\left(\frac{u_c}{\hat{u}}\right)_a$ is calculated as,

$$\left(\frac{u_c}{\hat{u}}\right)_a = \lambda_1 + \lambda_2 \frac{\hat{u}}{\sqrt{gd}} + \lambda_3 \exp\left(-\lambda_4 \frac{\hat{u}}{\sqrt{gd}}\right), \tag{6}$$

with $g$ is acceleration due to gravity and:

$$\lambda_1 = 0.5 - \lambda_3, \tag{7}$$

$$\lambda_2 = \lambda_3 \lambda_4 + \lambda_5, \tag{8}$$

$$\lambda_3 = \frac{0.5 - \lambda_5}{\lambda_4 - 1 + \exp(-\lambda_4)}, \tag{9}$$

$$\lambda_4 = \begin{cases} -15 + 1.35\left(T_s \sqrt{g/d}\right), & T_s \sqrt{g/d} \le 15 \\ -2.7 + 0.53\left(T_s \sqrt{g/d}\right), & T_s \sqrt{g/d} > 15 \end{cases}, \tag{10}$$

$$\lambda_5 = \begin{cases} 3.2 \cdot 10^{-3}\left(T_s \sqrt{g/d}\right)^2 + 8 \cdot 10^{-5}\left(T_s \sqrt{g/d}\right)^3, & T_s \sqrt{g/d} \le 20 \\ 5.6 \cdot 10^{-3}\left(T_s \sqrt{g/d}\right)^2 - 4 \cdot 10^{-5}\left(T_s \sqrt{g/d}\right)^3, & T_s \sqrt{g/d} > 20 \end{cases}. \tag{11}$$

In the present study, also based on the measurements from Egmond Beach, we found that the velocity skewness strongly depended on the Ursell number (Figure 5). The maximum skewness is determined as,

$$\left(\frac{u_c}{\hat{u}}\right)_{max} = 0.0235 \ \ln(U_r) + 0.552. \tag{12}$$

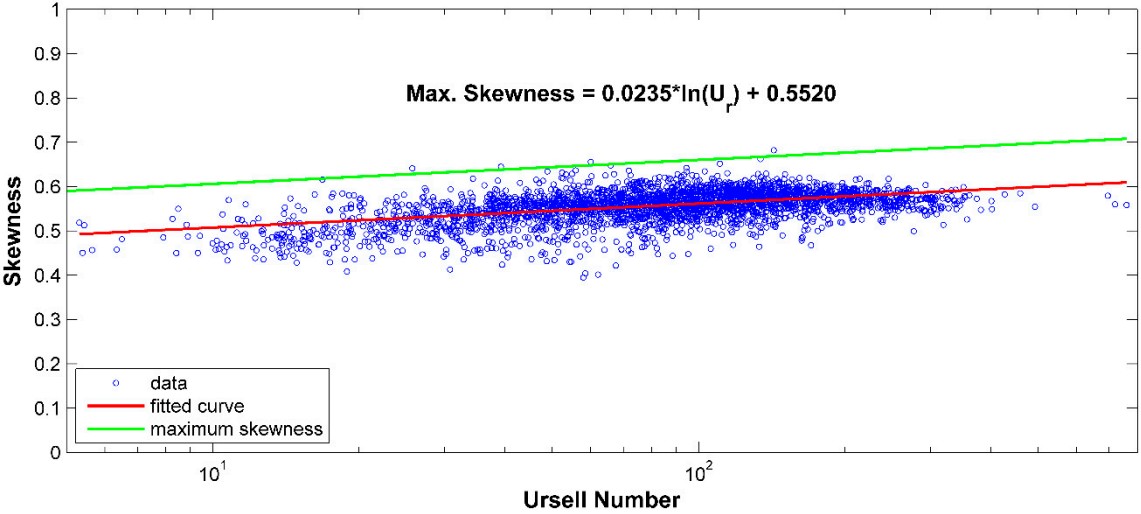

**Figure 5.** Relationship between velocity skewness and the Ursell number.

This expression is more general than the previous formulas [1,21], and can be applied for a wide range of wave conditions. For the Egmond Beach, the maximum skewness based on Equation (12) varied from 0.59 to 0.71.

The near-bed peak offshore orbital velocity ($u_t$) is then simply calculated from the full amplitude of the orbital velocity and the peak onshore orbital velocity according to,

$$u_t = \hat{u} - u_c. \tag{13}$$

## 4. Model Validations

*4.1. Validations Against Field Data Collected at Egmond Beach, The Netherlands*

Figure 6 shows the comparison between the calculated significant wave height and measurements at five locations E1–4 and E6 at Egmond Beach (see Figure 1 for locations). The blue solid line represents the calculations, whereas the red dot line describes the measurements. As can be seen, the simulated significant wave height agreed well with the measurements at all measurement locations. During the storm event on 25 October 1998, the calculations slightly underestimated the measurements at E1, E2 and E6. However, the model reproduced the measurements well for the remaining time of the field campaign.

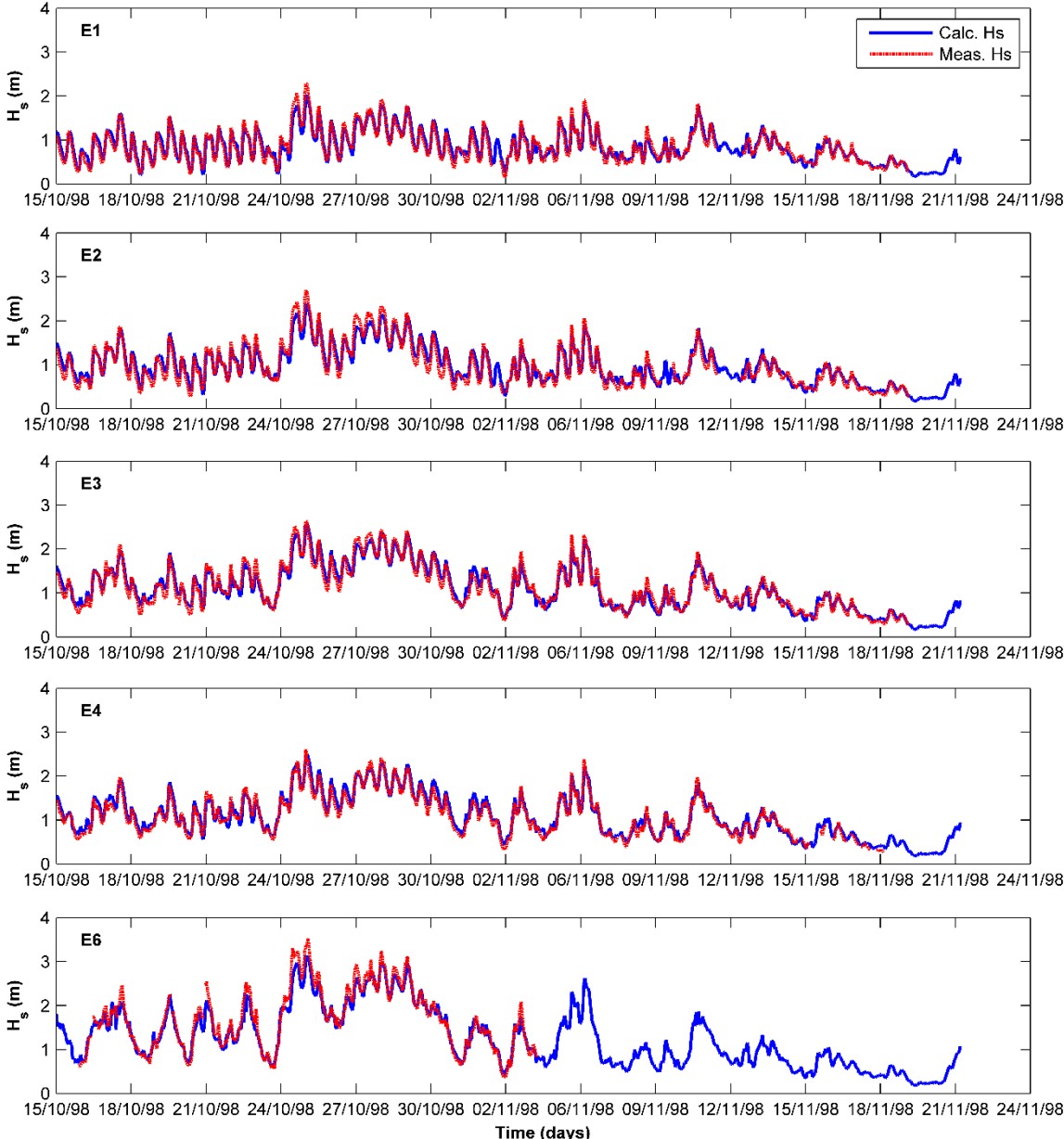

**Figure 6.** Calculated significant wave height and measurements at Egmond Beach.

In order to quantify the performance of the model, several indexes, including the relative root-mean-square error (*rel.rmse*), scatter index (*s.i*), relative bias (*rel.bias*), coefficient of determination ($R^2$), and Brier skill score (*BSS*), were employed [33,36,37]. For all data points at the measurement

locations, the *rel.rmse* was approximately 10.3%, the *rel.bias* was quite small 0.019, and the coefficient of determination $R^2$ about 0.95 and the *BSS* skill 0.94. Thus, the modified-EBED model provides reliable wave parameters needed for calculating the peak onshore and offshore orbital velocities and the velocity skewness.

The computed results for near-bed peak onshore orbital velocity at five locations E1–4, and E6 during the field campaign were assessed against measurements (Figure 7). The model simulations were in very good agreement with measurements at four nearshore measured locations (E1–E4). At location E1 the model slightly overestimated the measurements. The simulations were also higher than measurements at the end of October 1998 at locations E2, E3, and E4. On the other hand, at position E6 near the outer bar, where the peak onshore velocity observed was more than 2 m/s during the storm, the calculated peak onshore orbital velocity was significantly smaller than the measurements. The maximum discrepancy between calculation and measurement at location E6 during the storm was approximately 28%. The main reason for the discrepancy could be due to the underprediction of significant wave height obtained by the modified-EBED model during the storm, and the local wind factor was not considered in the model. However, during calm weather conditions, the calculation fits well with measurements at this location.

Figure 8 presents the comparison between the calculated near-bed peak offshore orbital velocity with measurements at the aforementioned five locations. In general, the predictions were in good agreement with measurements in shallow water. During the period between 25 October and 30 October 1998, the computed peak offshore velocity underestimated measurements at locations E3 and E4. As for the peak onshore velocity, the simulated results on peak offshore velocity were smaller than measurements at location E6. The maximum difference between calculation and measurement during the storm was approximately 42%.

The validation showed that the predictions by the numerical model agreed well with measurements. The *rel.rmse* for the peak onshore velocity was about 21.3%, whereas the *rel.bias* was quite small, approximately 0.013. The scatter index was 0.23, and both the coefficient of determination and the Brier skill score equaled 0.68. Although a satisfactory agreement between calculations and measurements was obtained, the calculated peak offshore velocity somewhat underestimated the measurements. Therefore, the relative bias obtained was negative, approximately −0.068. The *rel.rmse* was approximately 21.2% and the scatter index was about 0.225. The coefficient of determination was 0.64, implying that the prediction with the present formulas reproduced 64% of the measured variation in near-bed peak offshore velocity. The Brier skill score gave a value of 0.64, a bit smaller than for the peak onshore velocity, still indicating satisfactory performance of the proposed formulas.

Inter-comparisons between the results obtained by the present formulas, the original formulas by [1], called IH-82 hereafter, the modified formulas by [21], called GR-03 hereafter, and formulas by [24,25], called RA-12 hereafter, were also carried out. Figure 9 illustrates the inter-comparisons between computed onshore and offshore orbital velocities using four employed formulas and measurements at Egmond Beach. As can be seen, for IH-82 and RA-12 formulas, the deviations between calculations and measurements of the peak onshore orbital velocity were quite large. Furthermore, the computed peak offshore orbital velocity by the IH-82 and RA-12 were significantly underestimated measurements. The GR-03 produced better results, although the calculated peak onshore orbital velocity overestimated measurements.

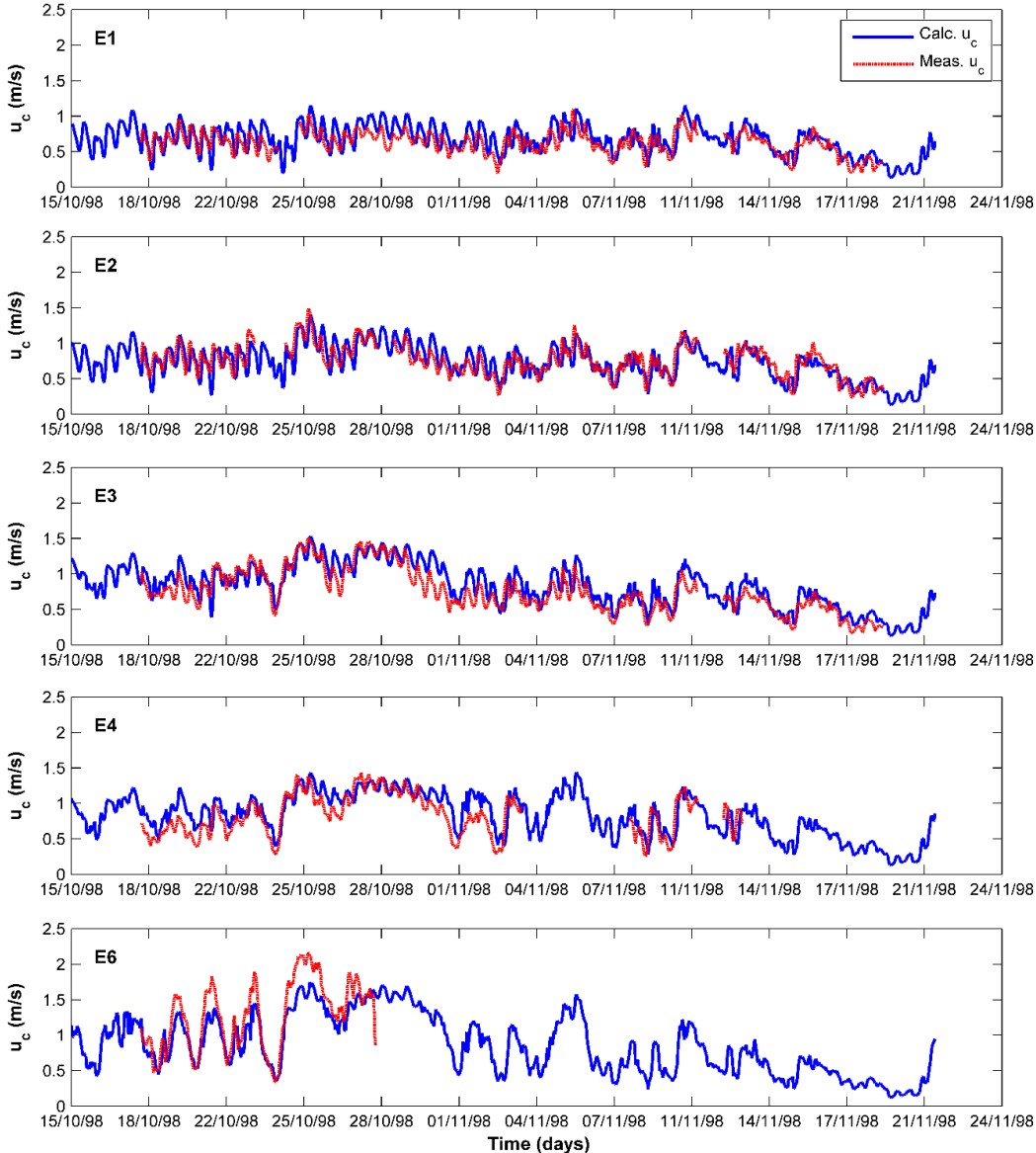

**Figure 7.** Calculated and measured peak onshore orbital velocity at Egmond Beach.

Table 3 summarizes the values on all the five quantitative indexes obtained with the present formulas as well as with the other formulas studied, for the field data from Egmond Beach. Based on the quantitative indexes, the present formulas reproduced the measurements well for both peak onshore and offshore orbital velocities.

Figure 10 presents the comparisons between the computed velocity skewness obtained by aforementioned formulas and measurements. The predicted velocity skewness using the proposed formula was more accurate than the previous formulas. The relative root-mean-square error, scatter index, and relative bias were the smallest, and equal to 11.81%, 0.12, and 0.05, respectively. The velocity skewness obtained by all four employed formulas somewhat overestimated the measurements. The variation in scatter was largest for IH-82, and the predictions by RA-12 almost all overestimated the measurements. The results for GR-03 were better than IH-82 and RA-12, but the number of scatter point located under the line of perfect equality was more than by the present formula.

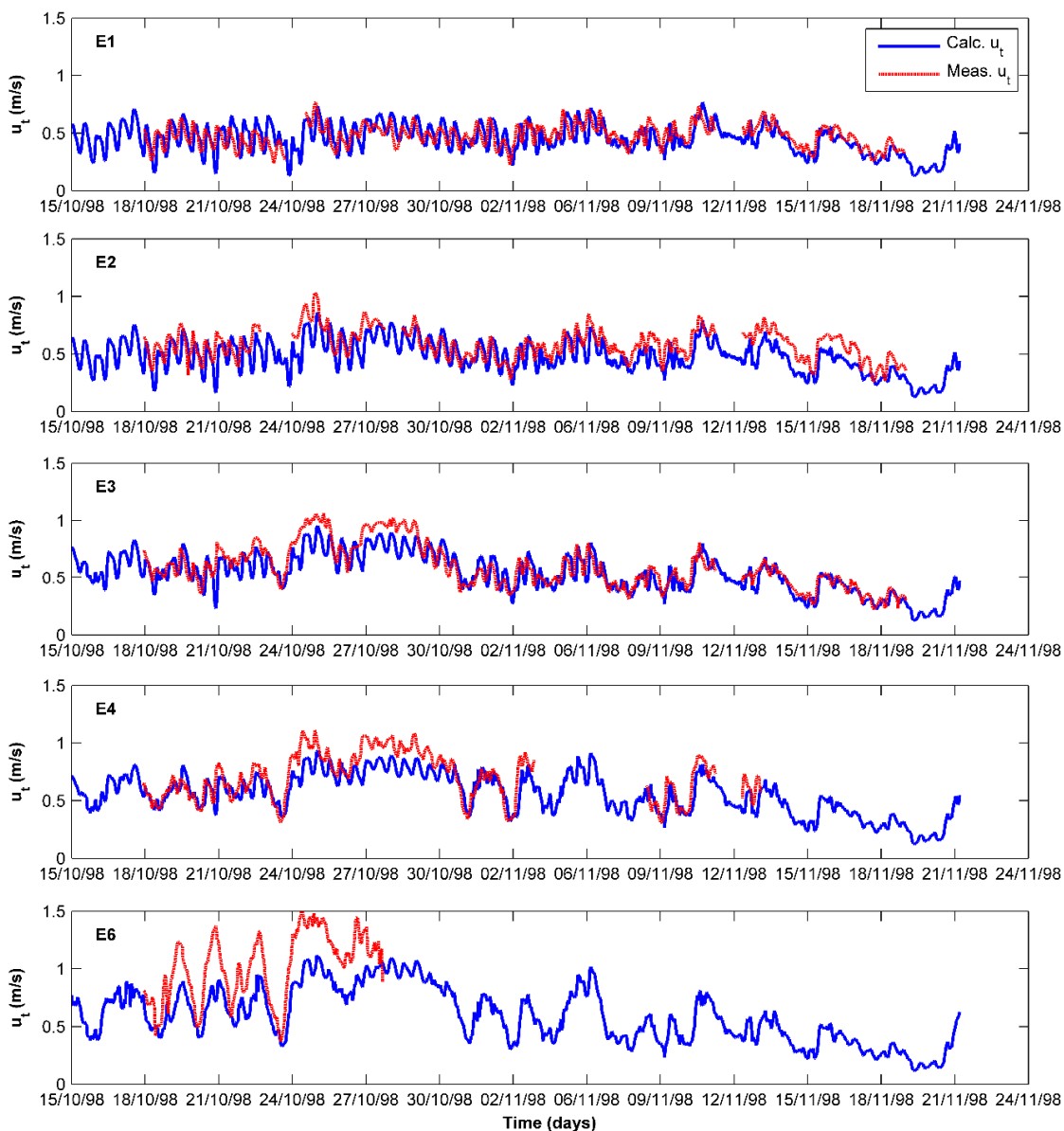

**Figure 8.** Calculated and measured peak offshore orbital velocity at Egmond Beach.

**Table 3.** Quantitative assessment of formula performances for the Egmond data.

| Formulas | *rel.rmse* | | *s.i* | | *rel.bias* | | $R^2$ | | *BSS* | |
|---|---|---|---|---|---|---|---|---|---|---|
| | $u_c$ | $u_t$ | $u_c$ | $u_t$ | $u_c$ | $u_t$ | $u_c$ | $u_t$ | $u_c$ | $u_t$ |
| Present | 21.32 | 21.19 | 0.230 | 0.225 | 0.013 | −0.068 | 0.68 | 0.64 | 0.68 | 0.64 |
| IH-82 | 27.04 | 39.75 | 0.291 | 0.422 | −0.051 | −0.337 | 0.51 | 0.54 | 0.48 | 0.49 |
| GR-03 | 26.02 | 22.91 | 0.282 | 0.234 | 0.129 | −0.105 | 0.65 | 0.63 | 0.64 | 0.62 |
| RA-12 | 30.85 | 41.40 | 0.333 | 0.440 | 0.181 | −0.366 | 0.65 | 0.61 | 0.53 | 0.54 |

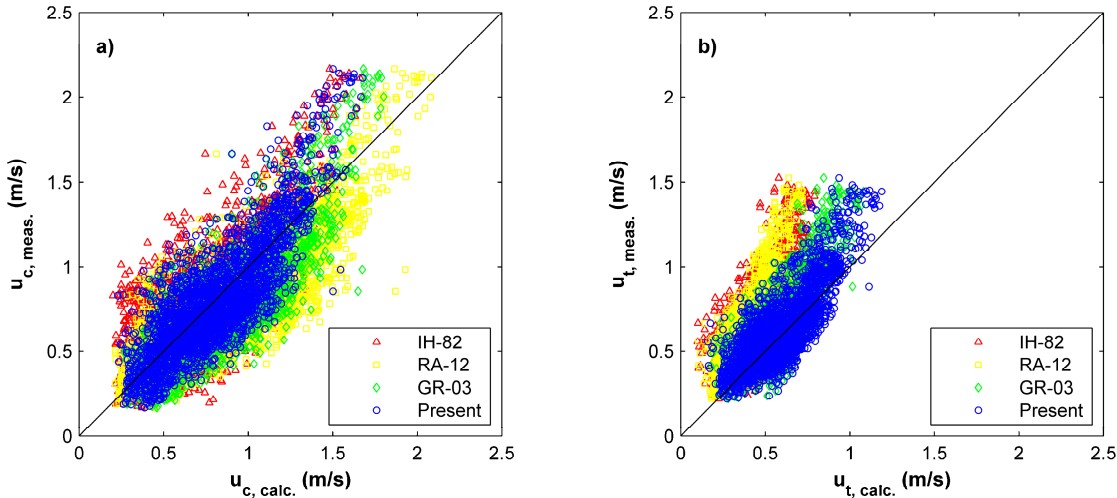

**Figure 9.** Comparison between computed peak onshore (**a**) and peak offshore (**b**) orbital velocity obtained by the studied formulas and measurements for Egmond Beach.

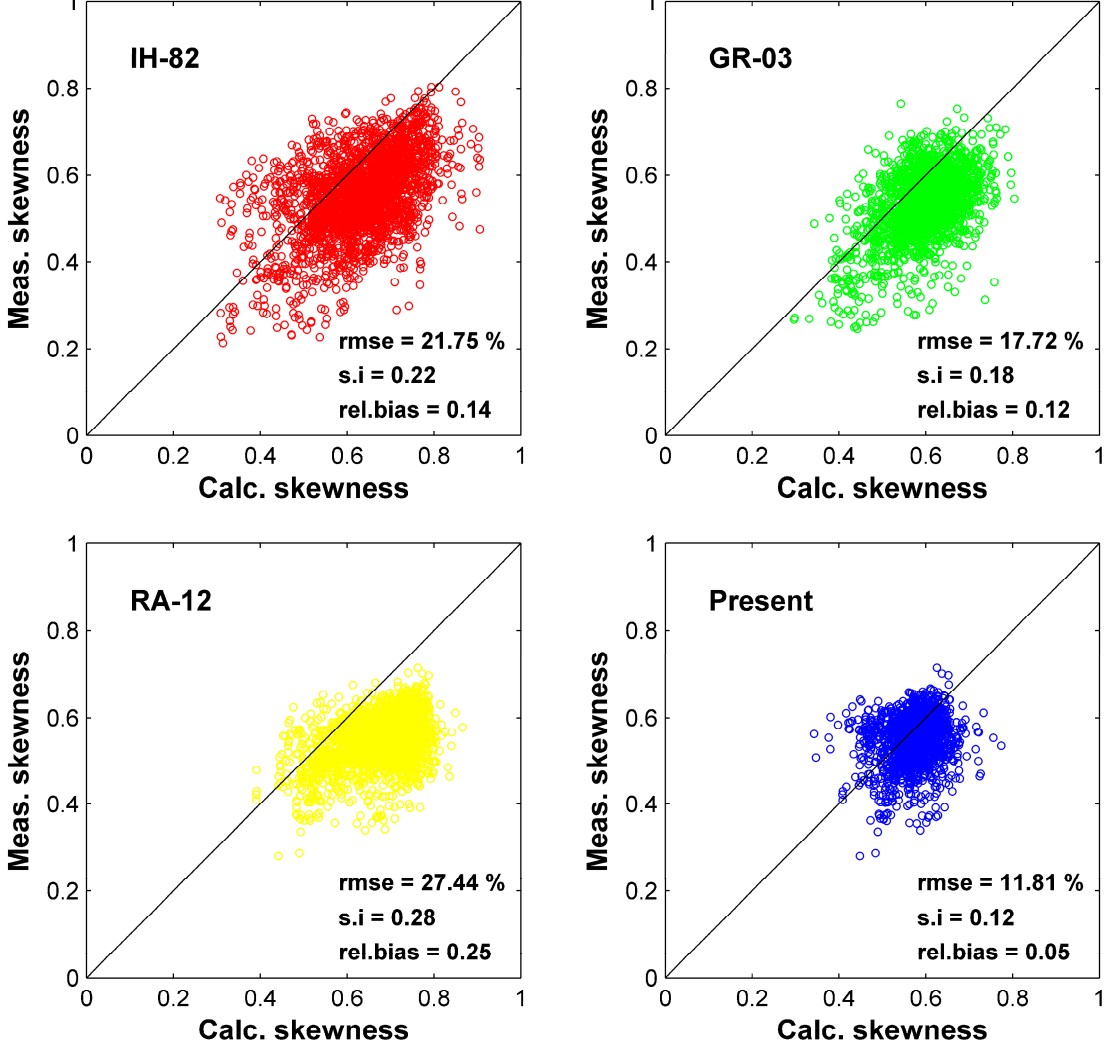

**Figure 10.** Predicted skewness using four employed formulas and measured skewness for Egmond Beach.

### 4.2. Validations against Small-Scale Laboratory Data from Delft University of Technology (GR99 Data)

Two data sets B1 and B2 from the GR99 data [29] were employed to verify the model. All calibrated parameters used for the Egmond case study were employed for test B1 and B2. Figure 11 shows the comparison between the computed results obtained by the proposed formula and the IH-82, GR-03, and RA-12 formulas and the measurements for both test B1 and B2. Predictions by all employed formulas somewhat underestimated measurements for both peak onshore and offshore orbital velocities. The RA-12 predicted well the peak onshore orbital velocity; however, it produced large underestimations for the peak offshore velocity, similar to the IH-12 and GR-03 formulas. Using the proposed formula, the best predictions for the peak onshore and offshore velocities were obtained, especially for the peak offshore orbital velocity. The quantity indexes for all four aforementioned formulas were listed in detail in Table 4. As can be seen, the relative rms error, relative bias, and scatter indexes for the proposed formula were quite small, implying that the present model could successfully reproduce the measurements of test B1 and B2.

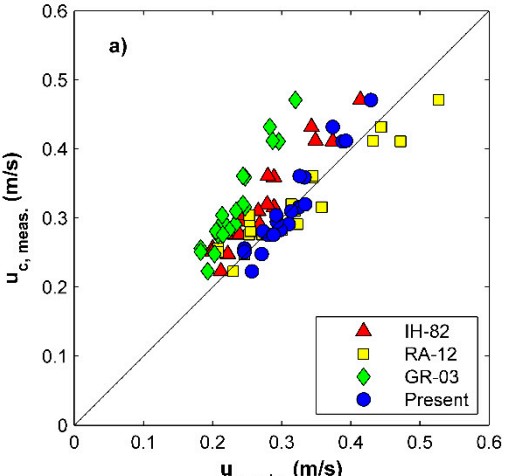 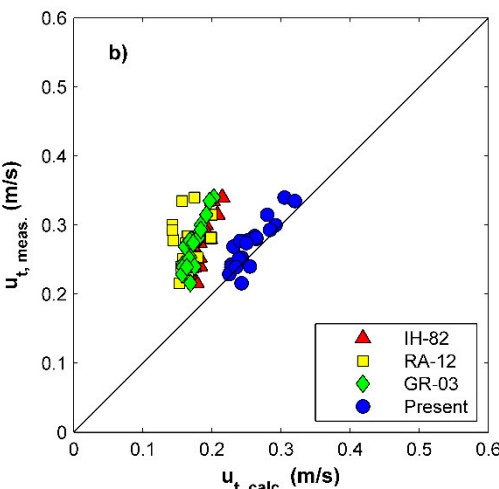

**Figure 11.** Comparison between computed peak onshore (**a**) and peak offshore (**b**) orbital velocity obtained by the studied formulas and measurements for test B1 and B2.

**Table 4.** Quantitative assessment of formula performances for the GR99 data.

| Formulas | rel.rmse | | s.i | | rel.bias | | $R^2$ | | BSS | |
|---|---|---|---|---|---|---|---|---|---|---|
| | $u_c$ | $u_t$ | $u_c$ | $u_t$ | $u_c$ | $u_t$ | $u_c$ | $u_t$ | $u_c$ | $u_t$ |
| Present | 7.20 | 7.94 | 0.073 | 0.080 | −0.017 | −0.054 | 0.93 | 0.78 | 0.88 | 0.77 |
| IH-82 | 15.89 | 32.77 | 0.162 | 0.330 | −0.150 | −0.319 | 0.92 | 0.73 | 0.91 | 0.52 |
| GR-03 | 27.88 | 36.44 | 0.285 | 0.367 | −0.266 | −0.358 | 0.92 | 0.78 | 0.76 | 0.53 |
| RA-12 | 9.89 | 40.18 | 0.101 | 0.405 | −0.003 | −0.383 | 0.91 | 0.01 | 0.76 | −0.15 |

### 4.3. Validations against Large-Scale Laboratory Data from Delta Flume, Delft Hydraulics (LIP11D Data)

The model was also validated against three data sets Test 1A–C from the large-scale Delta flume [30]. Overall, the predictions by the model for the significant wave height, peak onshore and peak offshore velocities agreed well with the measurements.

Figure 12 illustrates the comparison between computed results obtained by the four aforementioned formulas and the measurements from Test 1A–C of the LIP11D data. The predictions by IH-82 and GR-03 underestimated measurements for both peak onshore and offshore velocities. As for the small-scale data sets, the computed peak onshore velocity using RA-12 agreed well with measurements; however, the predicted peak offshore velocity underestimated the measurements, similar to the results obtained with IH-82 and GR-03. None of the four formulas could reproduce the measurement of peak

onshore orbital velocity at $x = 130$ m for Test 1C. Perhaps, this was an outlier in the measurements. Although the scatter was larger than the case with small-scale data sets, the predictions by the proposed formula were also better than the previous formulas, especially for the peak offshore orbital velocity. This is clear from Table 5 that summarizes all quantifying indexes for the data from Test 1A–C.

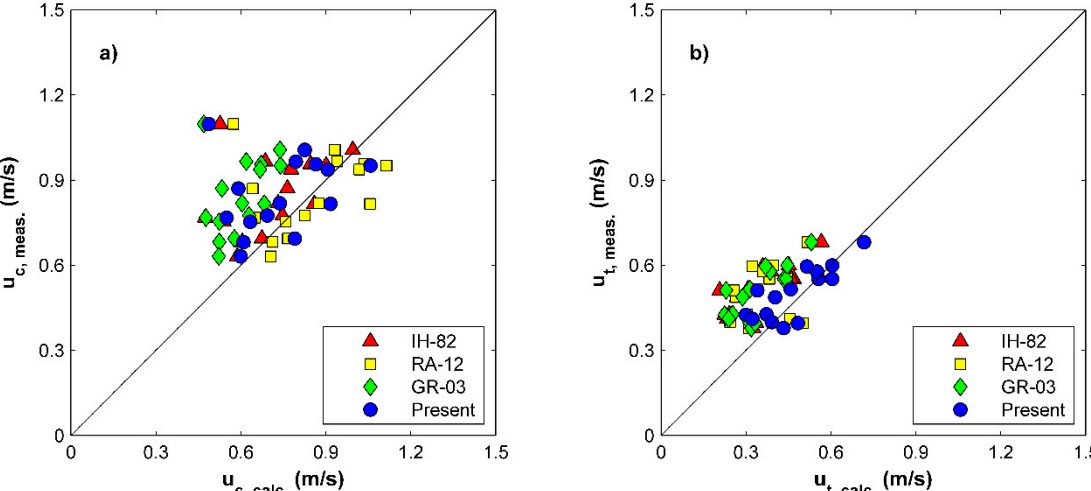

**Figure 12.** Comparison between computed peak onshore (**a**) and peak offshore (**b**) orbital velocity obtained by the studied formulas and measurements for Test 1A, 1B, and 1C.

**Table 5.** Quantitative assessment of formula performances for the LIP11D data.

| Formulas | rel.rmse | | s.i | | rel.bias | | $R^2$ | | BSS | |
|---|---|---|---|---|---|---|---|---|---|---|
| | $u_c$ | $u_t$ | $u_c$ | $u_t$ | $u_c$ | $u_t$ | $u_c$ | $u_t$ | $u_c$ | $u_t$ |
| Present | 24.01 | 15.08 | 0.243 | 0.153 | −0.131 | −0.062 | 0.07 | 0.64 | −0.78 | 0.37 |
| IH-82 | 23.46 | 33.93 | 0.237 | 0.345 | −0.158 | −0.314 | 0.17 | 0.53 | −0.33 | 0.36 |
| GR-03 | 32.59 | 33.52 | 0.330 | 0.340 | −0.294 | −0.319 | 0.14 | 0.59 | 0.06 | 0.54 |
| RA-12 | 20.44 | 34.83 | 0.207 | 0.354 | −0.010 | −0.290 | 0.10 | 0.13 | −0.82 | −0.30 |

The calculated velocity skewness obtained by the four employed formulas for both small and large-scale laboratory data was inter-compared and the result is presented in Figure 13. The red circle represents the comparisons for the GR99 data, whereas the blue triangle for the LIP11D data. As for the Egmond case study, the predictions by RA-12 and IH-82 significantly overestimated the measurements, for which the largest variation was obtained by the RA-12 formula. The results by GR-03 were improved and better than IH-82; however, they also overestimated the measurements. The predicted skewness by the proposed formula was in good agreement with the measurements, especially for the GR99 data. The underestimation of the skewness was mainly due to the underpredicted peak onshore orbital velocity for Test 1C of the LIP11D data. The quantitative indexes for the proposed formula were also better than previous formulas. The relative rms error, scatter index, and relative bias were the smallest, indicating the improvement of the present formula for calculating near-bed orbital velocity and skewness.

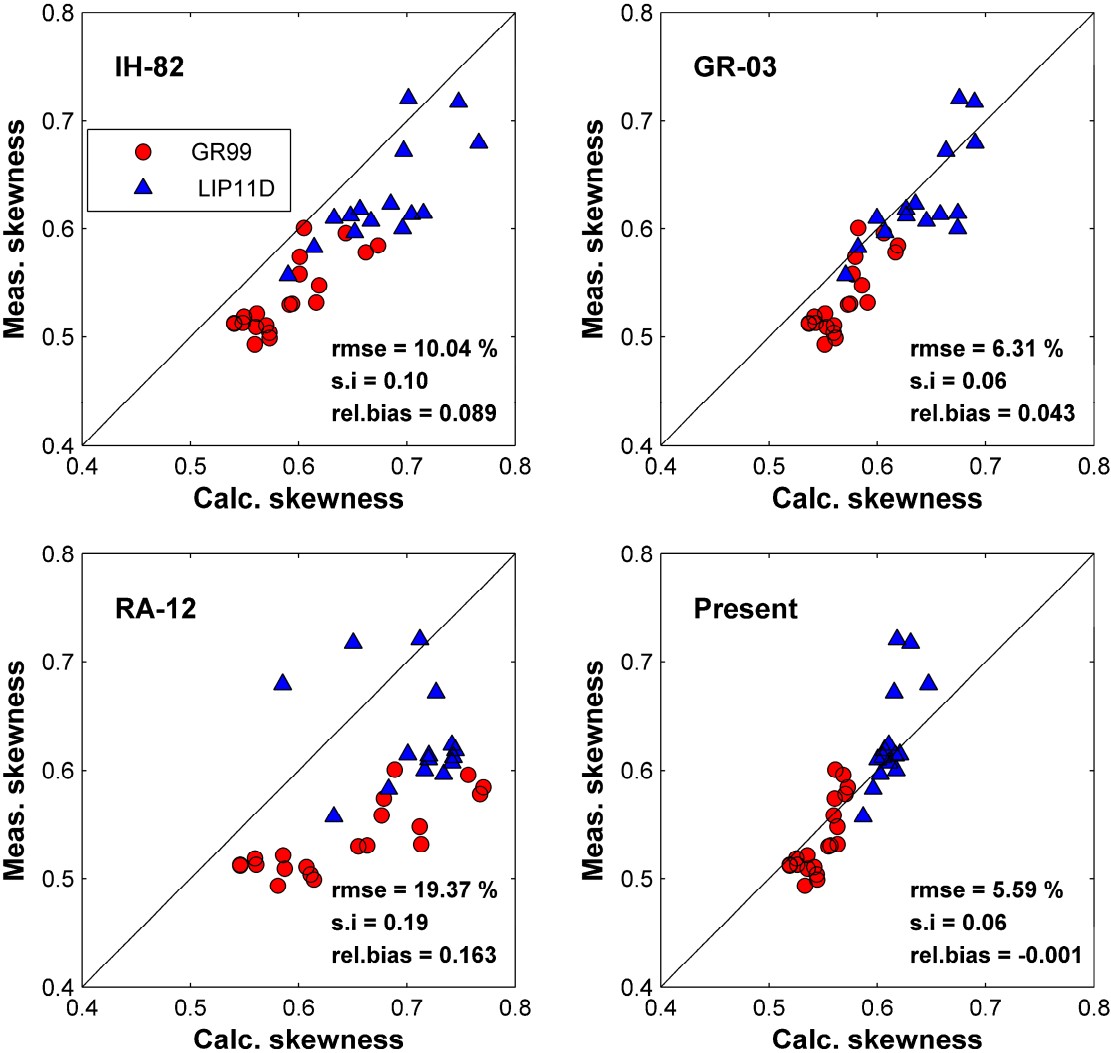

**Figure 13.** Predicted skewness using four employed formulas and measured skewness for GR99 and LIP11D data.

## 5. Discussion

The correction coefficient plays an important role in the formulas of IH-82. It directly affects the full amplitude of the near-bed orbital velocity. Consequently, the peak onshore, peak offshore orbital velocity, as well as velocity skewness are sensitive to change in the correction coefficient. In the original IH-82 formula, the correction parameter depends on the offshore wave conditions and water depth, and the obtained results often underestimate the measurements, especially for the offshore orbital velocity. It means that the computed full amplitude of the orbital velocity is underpredicted by the IH-82 formula. Although the prediction of peak onshore and offshore velocities was improved by GR-03 compared to the IH-82 formula, their prediction of velocity skewness still overestimated the measurements. In the present study, the correction parameter is determined based on the Ursell number. Predictions for both peak onshore and offshore orbital velocities were in good agreement with measurements from not only the field data at Egmond Beach but also the small and large-scale laboratory data. Thus, the estimation of the full amplitude of the orbital velocity was more accurate than the original IH-82 and GR-03 formulas. Furthermore, it is interesting to note the tendency of the correction coefficient to increase when the value of the Ursell number decreases to 5 from the field data at Egmond Beach. It could be because the measured full amplitude at location E6 was significantly larger than the full amplitude obtained by linear wave theory. Therefore, more field data in deeper

water are required to validate Equation (3) developed in the present study. It is surmised the correction parameter should be close to 1.0 when the Ursell number approaches zero.

The maximum skewness by the original IH-82 formula was simply calculated only on beach slope (Table 1). However, this expression is invalid for the beach with a very small slope ($\beta \to 0$). Furthermore, it cannot be applied for barred beaches, which include both positive and negative local slopes and where a representative value may be difficult to establish. The predicted velocity skewness by IH-82 formula was significantly large than the measurements for all investigated data. The GR-03 formula is an improvement on the original formulas. The maximum skewness depends on the local wave parameter and is constrained in the range between 0.62 and 0.75. However, this modification can produce overestimations of the velocity skewness for the field data collected at Egmond Beach, as well as the employed laboratory data. The modification of maximum skewness in the present study was improved and can be applied for different kinds of beach topography with a wide range of wave conditions.

Beside accurate predictions, consistency is an important factor for every semi-empirical formula that needs to be investigated for different beach topographies and wave conditions. Figure 14 shows a scatter diagram between the relative bias and the scatter index for both peak onshore and offshore orbital velocities calculated by the four aforementioned formulas and compared with all employed data sets. The IH-82 often underestimates the measurements, especially for the peak offshore velocity. The predictions using GR-03 overestimated measurements of the peak onshore orbital velocity for the Egmond Beach. Otherwise, predictions underestimated measurements for both peak onshore and offshore velocities. The RA-12 formula reproduced well the peak onshore velocity for GR99 and LIP11D data. However, its predictions of onshore orbital velocity significantly overestimated measurements for Egmond Beach, and dramatically underestimated measurements of offshore orbital velocity for all employed data. Although predictions by the proposed formula somewhat underestimated measurements, the consistency in the predicted results was clearly better than previous formulas.

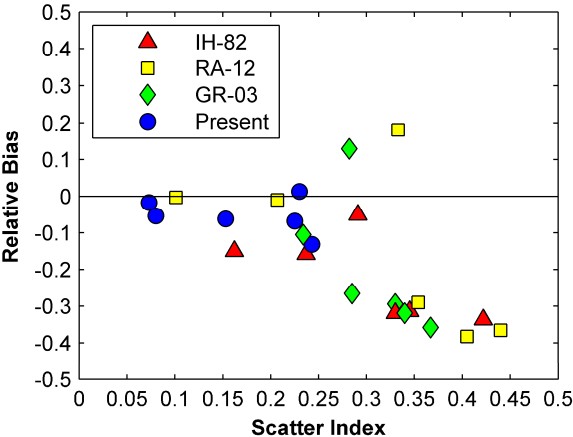

**Figure 14.** Scatter diagram between the relative bias and scatter index for the peak onshore and offshore orbital velocities using the four employed formulas.

The large discrepancies between calculations using RA-12 and measurements could be related to the total nonlinear parameter and the phase parameter in this formula [26]. The modification of [26] improved the prediction of the RA-12 formula for several data sets from a laboratory with a simple plane beach slope. However, the validation and application for natural beaches have not been assessed, especially for barred beaches such as Egmond or Duck, where the beach slope varies between a positive to negative value across the profile.

Finally, the local wave conditions, especially the wave height and period, are crucially important in the calculation of near-bed peak onshore and offshore orbital velocities. Accurate predictions of nearshore wave conditions will improve the predictions of the peak onshore and offshore orbital

velocity. All four employed formulas could not reproduce measurements at the position E6, Egmond Beach during the storm. The underestimation of the significant wave height at this position during the storm could be one of the reasons for the significant discrepancies between measured and calculated peak onshore and offshore orbital velocities. During the storm, the local wind could contribute to the increase in the wave height offshore. However, the wind factor was not included in the present model.

## 6. Conclusions

Following the general method of [1], modified formulations were proposed to calculate the near-bed peak offshore, onshore orbital velocities, and velocity skewness in shallow water. More specifically, the expressions for the correction coefficient and the maximum skewness were determined based on the Ursell number. The proposed formulas were validated against field data collected at the Egmond Beach and several laboratory data sets from the GR99 and LIP11D experiments. Previous formulations, including IH-82, GR-03, and RA-12, were also tested measurements.

The results obtained by the proposed formulations in the present study yielded improvements when compared to both the field and laboratory data. For the Egmond data, the relative rms errors for the peak onshore and offshore velocities were approximately 21.3% and 21.2%, respectively. The skill $R^2$ showed a value of 0.68 for the onshore velocity and 0.64 for the offshore velocity. For the GR99 data, the predictions agreed well with the measurements; the rms errors for the peak onshore and offshore velocities were only approximately 7.2% and 7.9%, respectively. For the LIP11D data, the rms error for the onshore orbital velocity was higher (about 24%) because of the underestimation for the Test 1C. However, the rms error for the offshore orbital velocity was only 15.1%. Consequently, the computed velocity skewness was significantly improved based on the accurate predictions of the onshore and offshore orbital velocities. The rms errors for the velocity skewness were approximately 5.6% for laboratory data and 11.8% for the field data. Overall, the predictions by the present formulations exhibited better agreement with the data than previously developed formulas.

**Author Contributions:** Conceptualization, P.T.N., J.S. and M.L.; methodology, P.T.N., J.S. and M.L.; software, P.T.N. and N.T.T.; validation, P.T.N. and N.T.T.; formal analysis, P.T.N.; investigation, P.T.N.; resources, J.S.; writing—original draft preparation, P.T.N. and N.T.T.; writing—review and editing, M.L., J.S. and P.T.N.; visualization, P.T.N. and N.T.T.; supervision, J.S. and M.L.; project administration, J.S.; funding acquisition, P.T.N. and J.S. All authors have read and agreed to the published version of the manuscript.

**Funding:** The work is partly funded by the Alexander von Humboldt Foundation in Germany, and partly funded by Vietnam National Foundation for Science and Technology Development (NAFOSTED) under grant number 107.03-2014.30.

**Acknowledgments:** The authors would like to thank Ruessink at Utrecht University, and Kleinhout at Port of Rotterdam for providing the data collected at Egmond Beach, The Netherlands. The authors would also like to thank Yamashiro at Kyushu University for providing the original paper of Isobe and Horikawa (1982) as well as for interesting and fruitful discussions. The helpful assistance and interesting discussions with Arno Behrens, Sebastian Grayek, Antonio Bonaduce, and Gayer Gerhard at Dept. Hydrodynamics and Data Assimilation, Institute of Coastal Research, HZG are much appreciated. The authors would also like to thank the anonymous reviewers for their valuable comments.

**Conflicts of Interest:** The authors declare no conflict of interest.

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
