# Peer review of "Improved Calculation of Nonlinear Near-Bed Wave Orbital Velocity in Shallow Water: Validation against Laboratory and Field Data"

_jmse, doi:10.3390/jmse8020081_

Round 1

Reviewer 1 Report

The work conducted by the authors is good and the resulting manuscript deserves publication.

I have to suggest acceptance with only very minor revisions.

Introduction:

I suggest to improve the introduction by citing the following references:

Conley, D. C., & Griffin Jr, J. G. (2004). Direct measurements of bed stress under swash in the field. Journal of Geophysical Research: Oceans109(C3).

Miles, J., Butt, T., & Russell, P. (2006). Swash zone sediment dynamics: A comparison of a dissipative and an intermediate beach. Marine Geology231(1-4), 181-200.

van der Werf, J. J., Schretlen, J. J., Ribberink, J. S., & O'Donoghue, T. (2009). Database of full-scale laboratory experiments on wave-driven sand transport processes. Coastal Engineering56(7), 726-732.

Vicinanza, D., Baldock, T., Contestabile, P., Alsina, J., Cáceres, I., Brocchini, M., Conley D., Lykke Andersen, T., Frigaard, P., Ciavola, P. (2011). Swash zone response under various wave regimes. Journal of Hydraulic Research49(sup1), 55-63.

Line 138. Please, provide a picture of the instrumented beach.

Line 142. The points E1-6 should represent six sections in which are installed pressure sensors and 2 EMF. Therefore, instead to use a circle, the right symbol should be a segment, indicating the measuring section.  

Line 189. Typo error in the formula (5)

Line 191. Typo error in the formula (6)

Reviewer 2 Report

Comments on "Improved calculation of nonlinear near-bed wave orbital velocity in shallow water: validation against laboratory and field data" by P. T. Nam and collaborators

The work introduces a modification of a previous expression to obtain the velocity skewness under random waves. Velocity skewness is an interesting parameter to understand onshore and offshore sediment transport processes in the nearshore wave-dominated areas.

Leaving aside the potential interest for the reader (Editor) i find little physical insight. I do not meaning that the work is not useful for this reason.

My main concerns are:

.- the manuscript requires a THOROUGH review of the edition (English, figures, including labels, symbols in equations, abbreviations, captions of the tables, ...); i found several mistakes/typos ("were was", ...)

.- for what i undertood, the novelty is modifying r and (u_c/ tilde(u))_max relative to IH82. However, the original expressions provided by IH82 for these two dimensionless variables are not given. Related to this, equation (1) seems unnecessary (what is theta? is that correct?); equation (5) has some symbols in chinse, and i do not know what "u" (without tilde) is.

.- i miss a clear explanation on how all the variables are obtained from Uw (wich, i understand, is obtained from Hs and Tp in the linear theory). Is there any iterative process?

.- figure 4: i found surprising that the p1 is significantly negative. Actually, the cloud of points does not look very nice: "r" seems not to be a function of "Ur" but also a function of many other factors. I would find it interesting to provide 95% confidence lines in the plot also, and emphasize that the fitting is far from good.

.- equation (12): please provide the expression proposed by IH82; and also for "r";

.- define s.i. (capitalize it as SI) and Brier skill score, and provide an interpretation of the errors.

.- distance to the bottom: what is the influence? Is it really negligible? Cite sources if so. The authors use data at different heights.

Reviewer 3 Report

Review report for theJMSE manuscript

Improved calculation of nonlinear near-bed wave orbital velocity in shallow water: validation against laboratory and field data

Pham Thanh Nam, Joanna Staneva, Nguyen Thi Thao and Magnus Larson

GENERAL COMMENTS

The manuscript describes a study reporting a parametrization for calculating the nonlinear near-bed wave orbital velocity in the shallow water. Starting from Isobe and Horikawa’s equation [1], the authors provide a correction coefficient and maximum skewness as function of the Ursell number in order to achieve the near-bed peak offshore and onshore orbital velocities.

The modified-EBED model was used in order to describe the nearshore random wave transformation. It was originally developed by Mase (2001) and subsequently modified by Nam et al. (2009), in order to achieve improved description of waves in the surf zone.

The obtained equations for nonlinear near-bed velocity were validated against measurement from field data (Egmond Beach) and laboratory data (small-scale wave flume experiments at Delft University of Technology and large-scale wave flume experiments at Delft Hydraulics).

The results obtained by the proposed formulations in the paper yielded improvements when compared to both the field and laboratory data.

This is an interesting study and a well-written paper. The expression of correction coefficient r (line 177) and maximum skewness of velocity (line 201) obtained from field data (Egmond Beach) is more general of the formulas reached in the previous works mentioned in the paper, making applicable for a wide range of wave conditions. In general, the predictions were in good agreement with measurements in shallow water, but since the wind factor was not included in the present model, the employed formulas could not reproduce accurately measurement at the position E6 (Fig. 2) where the local wind could contribute to the increase in the wind height offshore, as suggested by the author.

In Fig. 4, the agreement between log(Ur) and r looks poor; value of the coeff. of determination should be reported and the quality of the agreement should be discussed

SPECIFIC COMMENTS

Line 15                       There is a typo:  accurately

Line 134                      There is a typo: pressure gauges

Lines 134 – 136           The nearshore wave parameters were measured. Besides the pressure sensors also wave gauges were used ?

Line 164                      Agreement is missing

Lines 189 – 191           There are Chinese symbols above “û” (full amplitude of the horizontal water particle   velocity).

Line 215                      A storm event on 25 Nov. 1998 is not present in Figure 6

Lines 220 – 221           According to Figures 6 and 7, the measurement locations are 5, not 6.

Lines 229 – 230           There are two typos. The location of the adverb very “The model simulations were in very good agreement” and the repetition of the subject “The simulation”
